# Paramedics in Switzerland: A Mature Profession

**DOI:** 10.3390/ijerph19148429

**Published:** 2022-07-10

**Authors:** Thomas Schmutz, Youcef Guechi, Sandrine Denereaz, Florian Ozainne, Marc Nuoffer, Aris Exadaktylos, Vincent Ribordy

**Affiliations:** 1Service des urgences, hôpital fribourgeois HFR, chemin des Pensionnats, 1708 Fribourg, Switzerland; thomas.schmutz@h-fr.ch (T.S.); youcef.guechi@h-fr.ch (Y.G.); 2École supérieure d’ambulancier et de soins d’urgence Romande, En Budron C8, 1052 Le Mont-sur-Lausanne, Switzerland; s.denereaz@es-asur.ch; 3École supérieure de soins ambulanciers Genève, ch. des Bougeries 15, 1231 Conches, Switzerland; florian.ozainne@edu.ge.ch; 4Medi, Zentrum für medizinische Bildung, Max-Daetwyler-Platz 2, 3014 Bern, Switzerland; marc.nuoffer@medi.ch; 5Universitäre Notfallzentrum, Inselspital, Freiburgstrasse 16C, 3010 Bern, Switzerland; aristomenis.exadaktylos@insel.ch

**Keywords:** paramedic, prehospital, interprofessionality, emergency medicine

## Abstract

This paper describes how the profession of paramedics has evolved in Switzerland and takes the perspective of public health. Ambulance drivers play an important role in the health system, not only as a response to emergencies, but also by working in an interprofessional and interdisciplinary manner in response to other public health needs, such as home care, triage, telemedicine and interhospital transfers. This pre-hospital system is rapidly evolving and relies on the work of paramedics.

## 1. Introduction

Within Europe, the training, skills and role of a paramedic working in the field vary greatly from country to country. Switzerland’s development of the paramedic profession has been a gradual process. Once regarded as simple transporters, paramedics have now become one of the pillars of Switzerland’s extra-hospital emergency response chain. They now receive three years of formalised theoretical and practical training dedicated to the transportation and care of patients, which enables them to safely perform a wide range of emergency treatments as well as, most importantly, to authorise the use of medical therapies. We will now describe this experience, from the history of the profession in Switzerland to the ways in which it is taught and its role within the emergency response chain.

## 2. Definition of the Paramedic Profession: Registered Paramedic, Advanced Federal Diploma of Higher Education

The term “paramedic” is often used in Switzerland to describe emergency medical service (EMS) employees. This title is used in the national framework study plan [1]. It has its origins in North American culture, where it is the paramedics alone who initially intervene in a prehospital setting. In Switzerland, a “paramedic” is a healthcare provider who takes charge of a patient and then ensures their transportation to the hospital and to whom specified acts of medical care are delegated in the context of an emergency when a medical professional is not present. These acts of delegation are made possible by the fact that training is exclusively geared towards the care of pre-hospital patients. This entails an advanced level of expertise among paramedics in the management of life-threatening emergencies [2]. They therefore provide assessment, triage, delegated medical care and transportation of the patient. However, a prehospital emergency physician supports the management of complex cases [3]. A distinction should be made between the term “paramedic” and the adjective “paramedical”, as a paramedical profession is any profession that provides care in close collaboration with the physician.

## 3. History in Switzerland

The first ambulances appeared in Switzerland after the Second World War; their mission was to transport patients. During the 1950s, the number of road accidents increased and the Vaud section of the Touring Club Suisse (TCS) noted major failings in the prehospital chain of emergency response. It alerted the Council of State. In response to the conclusions of a study carried out by the commander of the Vaud gendarmerie, in 1958, the Council of State adopted a law regulating the establishment and financing of “official first aid centres in the event of an accident”, and thereby created the first official ambulance network [4].

The first ambulances were rudimentary and were associated with a particular garage or, less frequently, hospital. The training of paramedics was brief and limited to learning about first aid, with their principal responsibility being to transport the patient; they were referred to as “stretcher bearers”. Following the creation of the Inter-Association de Sauvetage (IAS) in 1962, a national umbrella organisation in the field of emergency response and care, the cantons adapted their arrangements for the handling of prehospital emergencies by gradually improving paramedics’ skills [5]. From 1988 onwards, the IAS laid the foundations for more professional training and oversaw its certification. At the same time, the cantons began to regulate emergency response services and made recognised professional training mandatory for those wishing to work as paramedics. A right to practice was therefore required for them to be allowed to carry out their work.

In 1994, the IAS proposed to the Conference of Cantonal Health Directors that the training concept be revised. It awarded the mandate to issue training requirements and to ensure the recognition of paramedic training to the Swiss Red Cross (SRC). Protocols for emergency measures were taught from 1995 onwards and, through the validation of specific training, paramedics were able to provide more advanced care.

The Swiss Paramedics Association (*Association Suisse des Ambulanciers* (ASA)) was founded in 1989 to represent their interests. In 1998, the SRC, which regulates the non-university health professions, carried out a fundamental restructuring of their training (three years) [6]. In doing so, it made paramedics subject to medical responsibility for any act of care, whereby the tasks delegated to the paramedic when no physician was present had to be recorded in writing by a doctor responsible for the particular ambulance service. As a result, the paramedic profession was recognized as a health profession.

From 1998 onwards, schools set up programmes in accordance with the SRC’s requirements. The Canton of Ticino was the first to provide such a course and, in French-speaking Switzerland, the CEPSPE in Geneva (centre d’enseignement de professions de la santé et de la petite enfance) launched the new training curriculum in 1999. Today, there are seven training centres for paramedics (Zurich, Lucerne, Zofingen, Bellinzona, Bern, Lausanne and Geneva).

A federal diploma for paramedics (Paramedic with College of Higher Vocational Education and Training Diploma) has been offered since 2004. The advanced level of this training allows tasks to be delegated as required in the course of the profession’s evolution [7]. The IAS has published guidelines for the recognition of these paramedic rescue services, in order to ensure their standardisation and, thus, quality. Today, there are around one hundred of these services, the majority of which are certified. They ensure that the entire territory is covered and play an essential role in the medical emergency response provided to the Swiss population (illustration 1b). In 2017, they responded to more than 1200 callouts per day. They employ 3700 people, including 2500 paramedics [8].

## 4. Training and Ambulance Crew Configuration

The IAS proposes a definition for ambulance crew recommendations. The team must comprise at least two professionals; their training must be recognised. There are two levels of qualification: the ambulance technician with a federal certificate (*technicien ambulancier* (TA) and the paramedic with a diploma (*ambulancier diplômé* (AD), allowing a skill and grade mix. The emergency physician is usually not included in the ambulance crew. Some Swiss German emergency response services still employ transport aides, but these are not recognised by the IAS and ceased to be integrated into the emergency response chain in 2015 [9].

## 5. Ambulance Technician with Federal Certificate

Upon completing training, the ambulance technician may transport patients and provide those in a stable state of health with basic life support (BLS). They can manage these interventions independently and under their own responsibility. In rescue and lifesaving situations, they assume the role of second-in-command and assist the paramedics with their tasks, who then take over responsibility for the intervention. The 1800 h of training are completed whilst working for an emergency response service. The training is concluded by a professional examination comprising a theoretical test and simulated situations. A holder of this certificate can then progress to the paramedic diploma after two more years in school (with part-time employment or by incorporating work placements).

## 6. Paramedic with Diploma

The objective of the training is to allow the paramedic, either independently or in collaboration with a physician, to provide prehospital care and to perform lifesaving techniques on patients of any age who are in distress, in crisis or at risk. They handle the intervention and are part of the lifesaving process. This training lasts three years (5400 h of training). The theoretical training (35–40% of the time) is complemented by practical training as part of a rescue service (40–50%) and specific practical work placements in related professions (10–25%): care of the elderly or disabled, anaesthesia, emergency service, medical emergency call centre, home help service and/or geriatrics. Specific placements (obstetrics, psychiatry, paediatrics and intensive care) are sometimes added in order to allow as many of the problems encountered in the field to be covered as possible. The three years culminate in an examination (dissertation or practice-oriented project, placement qualification and case-based practical examination with an examination interview). The title obtained places these professionals in the tertiary level of education, qualifying them to “carry out complex professional activities involving high-level responsibilities”. According to the ISCED International Standard Classification of Education, this level of training places them at level 6 (equivalent to a bachelor’s degree, with level 7 being a master’s degree and level 8 a doctorate).

They must then complete 40 hours of continuing professional development per year [10]. This is primarily based on existing training courses (Pre-hospital Trauma Life Support PHTLS, Advanced Cardiovascular Life Support ACLS, Advanced Medical Life Support AMLS, Pre-hospital Obstetric Emergency Training POETand Acute Psychiatric Emergencies APEX).

## 7. Role of a Paramedic

In Switzerland, medical calls are handled by medical regulators (paramedics, nurses) using keywords and protocols. Ambulances are dispatched at the request of the Emergency Medical Call Centre which can be contacted on number 144, with each case being assigned a level of priority of from one to three. The intervention can be primary (at the scene of the event) or secondary (between two care structures) (Table 1). 

One of the paramedic’s objectives is to transport the patient safely. They are responsible for driving the rescue vehicle. In the training curriculum, less than 5% of the time is allocated to acquiring driving skills. The paramedic who is active in the field assumes responsibility for the intervention; they are in charge of assessing, categorising and treating patients and directing them to a suitable technical facility. This makes efficient triage essential, both because of the different treatment pathways (cerebrovascular accident, infarction with elevation of the ST segment, serious trauma and cardiac arrest) and the disparity between hospitals’ technical facilities. The severity of the patient’s condition is categorised using the National Advisory Committee for Aeronautics (NACA) score, based on the most serious clinical situation encountered during the intervention (Table 2). Others scores or scales are currently used such as the Stroke score, G-FAST, Swiss emergency triage scale (SETS) and STEMI screening.

The paramedic does not make a medical diagnosis. Instead, they establish the problem and must take appropriate action based on clinical reasoning. All clinicians naturally use this cognitive process of clinical reasoning, which underlies all treatment decisions, even medically delegated ones. Clinical reasoning is the process of collecting, evaluating and using the available information to make a decision [11]. They must assess the patient’s condition in order to determine what action is needed and when (BLS emergency measures, and then protocol-based care). In fact, acquiring the foundations of clinical reasoning is a pedagogical challenge for schools with only three years to develop this complex skill. This is particularly relevant given the changing demands on prehospital care, e.g., when deciding not to transport a patient or to refer him or her to another specialist without going to a hospital. These emergency protocols, which have been taught since 1995, initially authorised the paramedic, following specific training, to use a defibrillator in semi-automatic mode after detecting cardiac arrest, insert a venous catheter for the administration of volume expander in the event of haemorrhagic shock, specific antidotes in loss of consciousness (such as flumazenil or naloxon) and pain killers (such as morphine) in cases of acute pain. Formerly known as “action guidelines”, these protocols have continued to evolve and the current training completed by paramedics now permits the delegation of care protocols including many drugs and technical care. These protocols are drawn up by committees of paramedics and validated by an advising physician of the rescue service, who has undergone training in emergency medicine that is recognized by the Swiss Society of Emergency and Rescue Medicine SSERM). They allow care to be provided safely, while also assuring valuable medical resources in the midst of a global shortage of emergency physicians. Nevertheless, these protocols cannot replace medical reasoning in all situations. Paramedics alone provide approximately 80–90% of prehospital care. On the other hand, in the cantons with the greatest degree of delegation, prehospital medicalisation (by ground system, such as *service mobile d’urgence et de réanimation* (SMUR) and/or by helicopter) allows inter-professional care. It still has its place in complementing paramedics’ skills and remains essential to airway management and complex cases [12]. In French-speaking Switzerland, there is a specific commission that writes algorithms for use in. Today, schools are working to standardise these algorithms at a national level. Like any other healthcare profession, paramedics are bound to maintain medical confidentiality. Following the intervention, they draw up a report, which must be passed on to the receiving department. Back at the station, they are responsible for invoicing, ongoing training and maintaining the equipment and the vehicle.

## 8. Outlooks and the Future

One of the main issues is the standardisation of protocols and practices across Switzerland. Disparities can sometimes even be found within a particular canton. “Rapid responder” models are also currently being tested. These allow experienced paramedics to be deployed in an emergency if the ambulance is unavailable. This “solo paramedic” initiates the assessment of the patient, establishes the severity of their condition and performs first aid while waiting for help.

Consideration is also being given to the possibility of developing this system in the future to more closely resemble that of England, with an increase in the level of training allowing the delegation of additional duties, such as the organisation of home care for patients who do not require hospitalisation. In Switzerland, postgraduate education is an interesting direction for the tertiary-type B sector, which would allow continued proximity to the professional environment. The other option would be to develop advanced (master’s) courses based on the model of universities of applied sciences. With respect to this point, the professional and the ASA must open up the debate and establish their position.

The profession is also evolving in another direction: scientific publication. Today, paramedics have access to university-type training courses such as Certificate, Diploma and Master of Advanced Study. This allows them to acquire the methodological basis for publishing. Indeed, in recent years, paramedics have published in scientific journals on various topics such as the evolution of non-emergency care requests, on technical aspects during resuscitation or the treatment of pain [13,14,15]. Finally, the schools have developed this aspect of methodology considerably during the initial training. Some students manage to create or be integrated into research protocols [16].

## 9. Conclusions

The paramedic profession in Switzerland has developed gradually. After completing three years of training, they can now handle most prehospital emergencies independently. The delegation of emergency care in the prehospital setting has become possible thanks to validated training, open communication between the parties involved, the investment of paramedics in the delegation process and close medical supervision. With support from the SMUR (Mobile Emergency and Resuscitation Services), they form the basis of a robust prehospital system. The future will involve working on the disparities between the different cantons and the standardisation of the system. The development and recognition of the profession will also come about through the creation of new working methods (telemedicine, home care, rapid responder) and interprofessionality. The profession is in a constant state of evolution.

## Figures and Tables

**Table 1 ijerph-19-08429-t001:** Priority levels of primary and secondary interventions.

P1	Immediate engagement, priority signals engaged, for a life-threatening intervention or for an event on the public highway or in a public place
P2	Engage without delay, priority signals engaged only if necessary for progression, for intervention without likelihood of life-threatening injury
P3	Commitment without priority signals on scheduled request or allowing a delay
S1	Transfer of a patient in a life-threatening condition
S2	Transfer of a patient not in a life-threatening condition, but whose departure cannot be delayed
S3	Transfer on scheduled request

**Table 2 ijerph-19-08429-t002:** NACA severity scale.

0: No injury or illness
1: Injury/disease without any need for acute medical care
2: Injury/disease requiring examination and therapy by a physician, but hospital admission is not indicated
3: Injury/disease without threat of life but requiring hospital admission
4: Injury/disease that can possibly lead to deterioration of vital signs
5: Injury/disease with acute threat to life
6: Injuries/diseases transported after successful resuscitation of vital signs
7: Lethal injuries or disease with resuscitation attempted (without transportation)
9: Non codable

## Data Availability

Not applicable.

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
