# Peer review of "Paramedics in Switzerland: A Mature Profession"

_ijerph, 2022, doi:10.3390/ijerph19148429_

Round 1
Reviewer 1 Report
Dear Authors,
your manuscript is an interesting paper about the paramedic profession in Switzerland, but I have some comments about it:
- please specify the information on the personnel in the ambulance: Is it always two people? or it could be more? Are doctors also included in the ambulance?
- You use the statement "... In Switzerland, a" paramedic "is first and foremost an ambulance driver ..." does it mean that every paramedic is a driver? And should this be the name of people who have completed 3 years of training and education?
- In the article I miss a description of medical rescue activities that paramedics can perform in Switzerland. Are all activities performed independently? Or are there advanced activities that must be performed under the supervision of a physician, as it is in many European countries?
- section References should be adapted to the requirements of the journal.
Author Response
Dear reviewer,
Please find here our revisions in the attachemant.
Kind regards

Reviewer 2 Report
This is a description of development of paramedic services in Switzerland. It can be used as reference for other countries that still developing their paramedic service.
It is a nice over view but rather simplified. It does not highlight what are the challenges faced, and is there anything that stands out from other countries.
Why is it called mature profession? What is it that the paramedics can performed now in order to be called this. What are the training involved during the 3 years of diploma?
Author Response
Dear reviewer,
Please find here our revisions in the attachement
Kind regards
